# Resolving mechanisms of immune-mediated disease in primary CD4 T cells

Christophe Bourges[1,2], Abigail F Groff[3], Oliver S Burren[1,2], Chiara Gerhardinger[3], Kaia Mattioli[3], Anna Hutchinson[4], Theodore Hu[1,2], Tanmay Anand[1,2], Madeline W Epping[1,2], Chris Wallace[1,3], Kenneth GC Smith[1,2], John L Rinn[3,5] & James C Lee[1,2,3,*]

## Abstract

Deriving mechanisms of immune-mediated disease from GWAS data remains a formidable challenge, with attempts to identify causal variants being frequently hampered by strong linkage disequilibrium. To determine whether causal variants could be identified from their functional effects, we adapted a massively parallel reporter assay for use in primary CD4 T cells, the cell type whose regulatory DNA is most enriched for immune-mediated disease SNPs. This enabled the effects of candidate SNPs to be examined in a relevant cellular context and generated testable hypotheses into disease mechanisms. To illustrate the power of this approach, we investigated a locus that has been linked to six immune-mediated diseases but cannot be fine-mapped. By studying the lead expression-modulating SNP, we uncovered an NF-κB-driven regulatory circuit which constrains T-cell activation through the dynamic formation of a super-enhancer that upregulates *TNFAIP3* (A20), a key NF-κB inhibitor. In activated T cells, this feedback circuit is disrupted—and super-enhancer formation prevented—by the risk variant at the lead SNP, leading to unrestrained T-cell activation via a molecular mechanism that appears to broadly predispose to human autoimmunity.

**Keywords** CD4 T cells; GWAS; MPRA; super-enhancer; TNFAIP3

**Subject Categories** Computational Biology; Genetics, Gene Therapy & Genetic Disease; Immunology

## Introduction

Hundreds of genetic loci have been implicated in autoimmune and inflammatory diseases, but the mechanisms by which these effect diseases remain largely unknown (Claussnitzer *et al*, 2020). An important first step in uncovering these mechanisms is to refine genetic associations down to specific causal variants, whose biological effects mediate disease risk, but statistical attempts to do this have been frustrated by linkage disequilibrium (LD), with only a minority of loci being resolved (Farh *et al*, 2015; Huang *et al*, 2017). Other methods have sought to re-weight candidate SNPs using their enrichment within functional genomic elements (e.g. tissue-specific regulatory marks; Schaub *et al*, 2012; Shooshtari *et al*, 2017), but these do not assess whether SNPs have functional consequences, nor reveal the biological effect that contributes to disease. This leaves the majority of GWAS loci either unresolved or unresolvable, and the ambition of identifying disease mechanisms largely unrealised (Visscher *et al*, 2017). To compound this challenge, the specific gene(s) that are affected by disease-associated variants have not been confirmed for most loci (Claussnitzer *et al*, 2020). Many associated haplotypes, for example, contain multiple genes with little evidence for any one being causally involved, while other associations are entirely located within intergenic regions (or "gene deserts") and are often reported to lack candidate genes.

Most GWAS associations are attributable to variation in regulatory rather than coding sequence, with significant enrichment in enhancers, and particularly super-enhancers—large enhancer clusters that are usually cell type-specific and control expression of key genes involved in cell state (Hnisz *et al*, 2013). Testing individual candidate SNPs for effects on transcription—as a means of refining disease-associated haplotypes to specific functional variants—would bypass the limitations of LD and directly assay the process that mediates disease risk, but was previously laborious and expensive. The development of high-throughput assays of enhancer activity, such as massively parallel reporter assays (MPRAs), has now made this possible. MPRAs simultaneously test the regulatory activity of large numbers of short sequences by coupling each to a barcoded reporter gene (Melnikov *et al*, 2012). By normalising the RNA barcode counts from transfected cells to their equivalent counts in the input plasmid library, MPRAs have identified genetic variants that modulate expression in various settings (Tewhey *et al*, 2016;

1 Cambridge Institute of Therapeutic Immunology and Infectious Disease, Jeffrey Cheah Biomedical Centre, Cambridge Biomedical Campus, University of Cambridge, Cambridge, UK
2 Department of Medicine, University of Cambridge School of Clinical Medicine, Addenbrooke's Hospital, Cambridge, UK
3 Department of Stem Cell and Regenerative Biology, Harvard University, Cambridge, MA, USA
4 MRC Biostatistics Unit, Cambridge Institute of Public Health, Cambridge, UK
5 Department of Biochemistry, BioFrontiers Institute, University of Colorado, Boulder, CO, USA
*Corresponding author. Tel: +44 1223 767063; E-mail: jcl65@cam.ac.uk

Ulirsch et al, 2016). A key feature of MPRAs, however, is that the results are determined by the transcription factors present within the transfected cells, and so could be misleading if an inappropriate cell type was used. To date, almost all MPRA studies have been performed in cell lines, in part because these are easy to culture and transfect. It is widely recognised, however, that these are poor surrogates for the types of the immune cells that drive autoimmune disease (Astoul et al, 2001; Bartelt et al, 2009).

Here, we adapt an MPRA for use in resting and stimulated primary CD4 T cells—the cell type whose regulatory DNA is most enriched for immune-mediated disease SNPs (Farh et al, 2015). By simultaneously testing candidate SNPs from 14 immune disease-associated gene deserts for expression-modulating activity, we generate a basis for exploring the underlying biology that can yield previously unappreciated insights into the effects of disease-associated variation. At the pleiotropic 6q23 locus, for example we uncover a molecular mechanism whereby a common variant—identified via its expression-modulating effect in CD4 T cells—disrupts an NF-κB-driven pathway that normally limits T-cell activation through the dynamic formation of a TNFAIP3 super-enhancer. Disruption of this feedback circuit releases activated CD4 T cells from an intrinsic molecular brake and reveals a mechanism by which a multi-disease-associated haplotype can causally change biology, and a pathway that would appear to be pervasively involved in human autoimmune disease.

## Results

### Adaptation of MPRA for use in primary CD4 T cells

To determine whether causal variants could be identified via their functional effects, we designed an MPRA to assess candidate SNPs (all variants with $r^2 \geq 0.8$ with the lead SNP) at 14 gene deserts linked to 10 immune-mediated diseases (Fig 1A, Table 1; Trynka et al, 2011; Cooper et al, 2012; Eyre et al, 2012; Jostins et al, 2012; Tsoi et al, 2012; International Genetics of Ankylosing Spondylitis C et al, 2013; International Multiple Sclerosis Genetics C et al, 2013; Liu et al, 2013; Onengut-Gumuscu et al, 2015). Several of these loci cannot be resolved by fine-mapping (Farh et al, 2015; Huang et al, 2017). Gene deserts were selected because (i) less is known about how these predispose to disease compared with regions containing candidate genes, (ii) other non-coding mechanisms (such as splicing effects) are unlikely to account for these associations, and (iii) many of these contain epigenetic marks consistent with enhancer activity (Hnisz et al, 2013). To maximise the genomic context tested around each SNP, we designed three overlapping constructs for every SNP allele (Ulirsch et al, 2016) and synthesised extra oligonucleotides to test combinations of risk alleles if more than one SNP could be assayed in the same construct. We also included oligonucleotides that tiled each locus at 50 bp intervals to test for enhancer activity—and enable us to exclude regions that lacked this.

After assembly, the MPRA plasmid library was transfected into primary CD4 T cells from healthy donors (Fig 1A, Materials and Methods) but no expression of the reporter gene was detected (Appendix Fig S1A and B). After confirming that successful transfection had occurred (Appendix Fig S1B), we surmised that the minimal promoter, which is conventionally used in MPRA, may be

insufficient to initiate transcription in primary T cells. In cell line-based MPRA studies, stronger promoters have been shown to produce highly comparable results to those obtained using a minimal promoter (Ernst et al, 2016; Ferreira et al, 2016). We therefore screened a series of promoters in CD4 T cells (Appendix Fig S1C) and selected the Rous sarcoma virus (RSV) promoter for incorporation into an adapted MPRA vector (Fig 1B) as this robustly initiated transcription but was not so strong as to preclude further amplification.

After assembly, the adapted MPRA plasmid library was transfected into resting and stimulated CD4 T cells from 12 healthy donors. Multiple biological replicates (donors) were used to ensure that the results were reproducible, and control for differences in CD4 T cell composition between donors and the reduced dynamic range expected with a stronger promoter. After 24 h, GFP was detected and RNA was extracted to quantify barcode expression by high-throughput sequencing (Appendix Fig S1D). After pre-processing, barcode counts were collapsed to individual genomic constructs for analysis (Materials and Methods). Using principal component analysis, we found that the activation state of T cells was responsible for much of the total variance (Fig 1C) and that the transcriptional activity of constructs—calculated by normalising RNA barcode counts to their respective counts in the plasmid library—correlated well between individuals (Fig 1D). To detect expression-modulating variants, we used QuASAR-MPRA (Kalita et al, 2018; Fig 2A) and combined the results from each donor using a fixed-effects meta-analysis (Materials and Methods). Significant expression-modulating activity was detected at one or more constructs for 8/10 positive control SNPs (comprising five expression-modulating variants (Tewhey et al, 2016), two single variant eQTLs (Farh et al, 2015) and three synthetic SNPs that included/disrupted a canonical transcription factor binding site; Fig 2B). Enhancer activity was also detected in the positive control regions for the tiling analysis, while no such activity was detected in the negative controls (Appendix Table S1, Materials and Methods). The effects observed in resting and stimulated CD4 T cells were highly correlated (Appendix Fig S1E), but these effects did not correlate particularly well with results obtained in Jurkat cells (an immortalised CD4 T-cell line)—reinforcing the value of using an appropriate cellular model when studying human disease (Appendix Fig S1F). To validate the observed effects, we tested the most significant expression-modulating SNP at each haplotype, two positive controls and five SNPs with no allele-specific effects using a complementary luciferase-based system (Fig 2C). Despite using a different promoter and quantification method, the MPRA and luciferase results were highly correlated (Pearson $r = 0.87$)—indicative of genuine expression-modulating effects that are likely to be physiologically relevant (Fig 2D). To determine whether these variants could have been prioritised by other means, we compared the MPRA results with in silico methods designed to identify functional variants—DeepSEA (Zhou & Troyanskaya, 2015) and RegulomeDB (Dong & Boyle, 2019; Dataset EV1). Considering these approaches together, the lead MPRA SNP was predicted to be the most functionally significant variant at 3/14 loci (2 by DeepSEA, 1 by RegulomeDB). DeepSEA also predicted that 3 more lead SNPs would be functionally significant, but prioritised other candidate SNPs at these loci (most of which had no expression-modulating effect in CD4 T cells). At the remaining eight loci, the lead MPRA SNP was

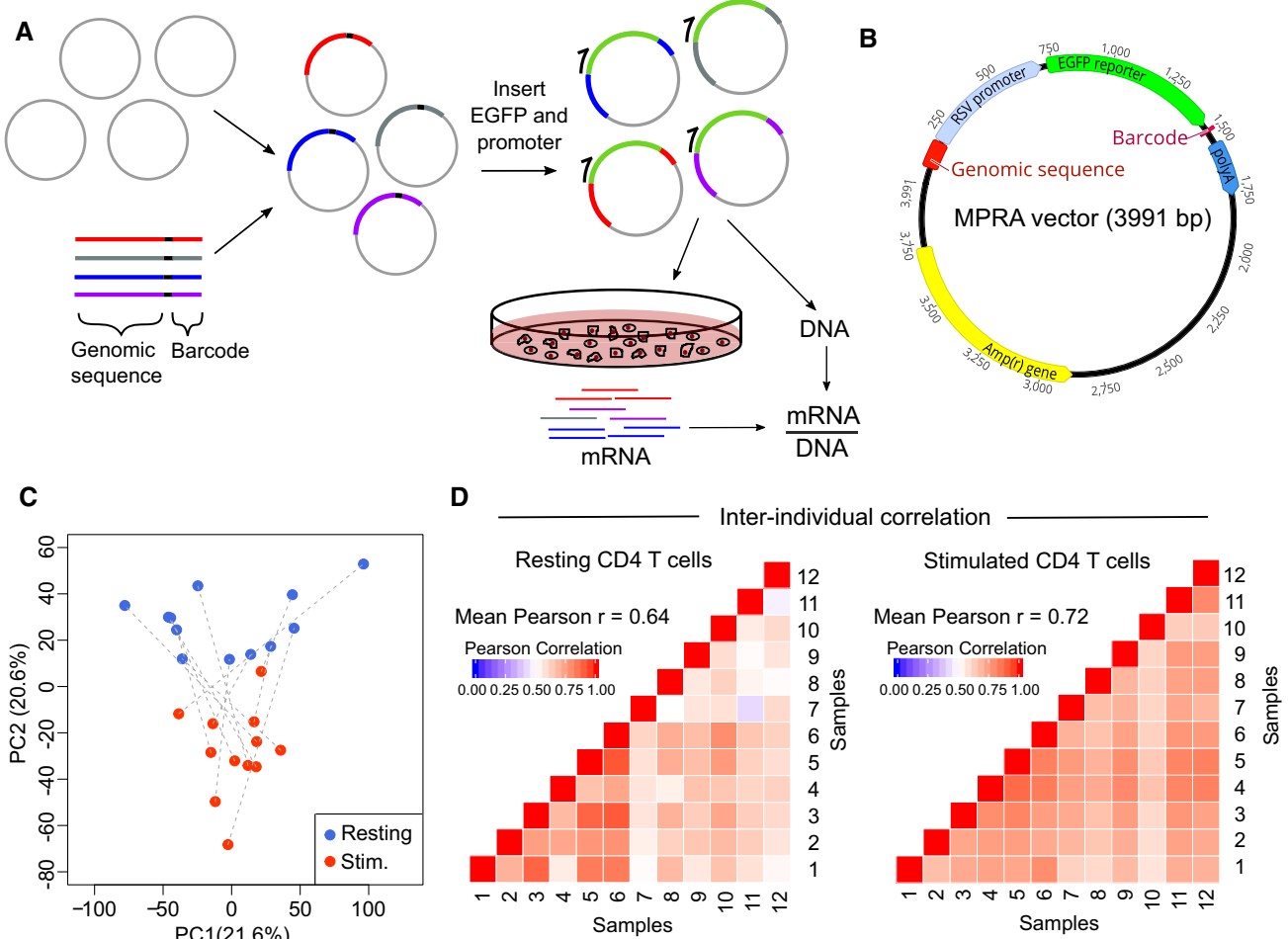

**Figure 1.   Development of MPRA for use in primary human CD4 T cells.**

A   Experimental workflow for MPRA: oligonucleotide library is cloned into an empty vector, and a reporter gene and promoter are inserted using restriction sites within the oligonucleotide. The assembled plasmid is transfected into primary CD4 T cells, and RNA is extracted after 24 h. RNA barcode counts are normalised to their respective counts in the input plasmid library (DNA), which is sequenced separately.

B   Adapted MPRA plasmid incorporating RSV promoter.

C   Principal component analysis of scaled element counts (sum of barcodes tagging same genomic construct in mRNA) in resting and stimulated CD4 T cells from 12 donors. Dotted lines indicate samples from the same donor.

D   Heat maps showing pairwise comparison of MPRA activity for all constructs (mRNA/DNA) between donors—left panel: resting CD4 T cells; right panel: stimulated CD4 T cells.

not predicted to have an expression-modulating effect—consistent with these methods being better at predicting negative effects than positive effects (Dong & Boyle, 2019) and highlighting the value of studying disease-associated loci in relevant primary cells. Altogether, these data indicate that MPRA can be adapted for use in primary CD4 T cells and that the results reflect the activation state of the cells and can identify constructs with regulatory effects.

**Adapted MPRA in CD4 T cells provides insights into the biological effects of genetic associations**

After establishing that MPRA could be performed in primary CD4 T cells, we next examined the results at disease-associated loci (Appendix Figs S2 and S3, Datasets EV2 and EV3). To assess

whether adapted MPRA would identify known causal variants, we used an inflammatory bowel disease (IBD)-associated locus that has been fine-mapped to a single variant, rs1736137 (Huang *et al*, 2017). In both resting and stimulated T cells, this SNP had highly significant expression-modulating activity, with the IBD-risk allele consistently increasing transcription (Fig 3A). As a further proof of principle, we next examined an ankylosing spondylitis-associated locus, where Bayesian fine-mapping using corrected coverage estimates resolves the association to three SNPs in the 99% credible set (rs6759298, rs4672505 and rs13001372). In stimulated T cells, one of these SNPs (rs6759298) had the most significant expression-modulating effect at this locus, while the others had negligible effects on transcription (Fig 3B)—thus resolving the likely causal variant.

**Table 1.   Autoimmune disease associations at 14 gene deserts.**

| tag SNP | $r^2$ between tag SNPs | Chr. | Associated disease(s) | Distance to nearest gene (kb) | Number of SNPs ($r^2 \geq 0.8$) | Haplotype block size (kb) |
|---|---|---|---|---|---|---|
| rs883220 | – | 1p34 | RA | 102.3 | 9 | 30.2 |
| rs6759298 (AS) rs10865331 (Ps) | 0.86 | 2p15 | AS, Ps | 99.5 | 12 | 33.9 |
| rs1534422 | – | 2p24 | ATD | 208.2 | 12 | 15.9 |
| rs1813375 | – | 3p24 | MS | 203.9 | 15 | 10.9 |
| rs2611215 | – | 4q32 | T1D | 139.4 | 20 | 16.7 |
| rs12186979 | – | 5p13 | AS | 59.7 | 5 | 97.8 |
| rs17119 | – | 6p23 | UC, CD, MS | 512.1 | 44 | 22.6 |
| rs2327832 (SLE) rs6920220 (rest)[a] | 0.92 | 6q23 | RA, CeD, UC, CD, SLE[b], T1D[c] | 143.6 | 9 | 47.5 |
| rs1991866 | – | 8q24 | UC, CD | 136.2 | 28 | 22.0 |
| rs2456449 | – | 8q24 | MS | 219.3 | 17 | 19.5 |
| rs4409785 | – | 11q21 | ATD, vitiligo[d] | 181.2 | 3 | 9.7 |
| rs1456988 | – | 14q32 | T1D | 1,086.9 | 38 | 14.0 |
| rs1297258 | – | 21q21 | UC, CD | 274.0 | 38 | 24.1 |
| rs2836883 (AS, PSC) rs2836878 (UC, CD) | 1.0 | 21q22 | AS, PSC, UC, CD | 87.1 | 13 | 5.8 |

AS, ankylosing spondylitis; ATD, autoimmune thyroid disease; CeD, coeliac disease; CD, Crohn's disease; Chr., chromosome; MS, multiple sclerosis; Ps, psoriasis; PSC, primary sclerosing cholangitis; RA, rheumatoid arthritis; SLE, systemic lupus erythematosus; SNP, single nucleotide polymorphism; T1D, type 1 diabetes; UC, ulcerative colitis.
Genetic associations were identified from published immunochip data. For each locus, the extended haplotype (LD region tagged by all SNPs with $r^2 \geq 0.8$ and extended by 50 bp on either side) does not contain coding or well-characterised non-coding genes. Haplotype block size represents region tagged by all SNPs with $r^2 \geq 0.8$.
[a]CeD tag SNP rs17264332 ($r^2 = 1.0$ with rs6920220).
[b]Association reported subsequently (Langefeld et al, 2017).
[c]Association reported using $P < 1 \times 10^{-5}$ to obtain a Bayesian posterior probability for T1D association given known associations with other diseases (Onengut-Gumuscu et al, 2015).
[d]Vitiligo association reported in GWAS, not immunochip (Jin et al, 2012).

We next investigated whether adapted MPRA could resolve possible causal variants at other loci, and so provide testable hypotheses into disease mechanisms. SNPs with strong functional effects were identified at several loci (Appendix Figs S2 and S3), including—for example—a chromosome six locus associated with both IBD and multiple sclerosis. Of 44 candidate SNPs in the shared risk haplotype, a single variant (rs34421390) had by far the largest and most significant expression-modulating effect in both resting and stimulated CD4 T cells (Fig 3C). This provides a focus for studying the upstream biology and demonstrates that the risk haplotype reduces transcription—an important finding since the locus interacts with the promoter of *JARID2*, a component of the polycomb repressive complex 2, in CD4 T cells (Javierre et al, 2016).

We made similar insights at a type 1 diabetes-associated locus, which contains 38 SNPs in strong LD. At this haplotype, the largest and most significant expression-modulating effect occurred with a construct containing the risk alleles for two adjacent SNPs (rs1988588 and rs3902659) which are 60 bp apart (Fig 3D). Both SNPs had similar effects when tested individually (with the risk allele reducing transcription) but these were weaker than with the construct containing both risk alleles (Fig 3D). This raises the possibility that the functional effect of this haplotype is mediated by a synergistic interaction between two adjacent SNPs, rather than a single causal variant—a prospect that could not be derived from genetic data since the SNPs are in complete LD ($r^2 > 0.9999$, Onengut-Gumuscu et al, 2015). This provides a basis to study the molecular mechanisms at this locus, which could help resolve the underlying biology.

Altogether, these results show that MPRA in primary CD4 T cells can identify SNPs that causally alter gene expression, and so provide testable hypotheses into disease mechanisms, while simultaneously identifying the nature of the functional effect.

**MPRA identifies an expression-modulating variant that disrupts NF-κB binding and super-enhancer formation**

To confirm that MPRA in primary CD4 T cells could help resolve disease mechanisms, we selected a pleiotropic locus on chromosome 6 for further study (Fig 4A). This region was chosen for several reasons. First, it was the only haplotype that was associated with six different diseases (Table 1), highlighting the biological importance of the locus. Second, despite receiving considerable attention, there is still uncertainty regarding the causal gene, with some studies implicating *TNFAIP3*, mainly because this is the closest plausible candidate (Jostins et al, 2012; Onengut-Gumuscu et al, 2015; Calderon et al, 2019) while others suggest that *IL20RA* is responsible (McGovern et al, 2016; Wu et al, 2019). Third, statistical fine-mapping has been attempted at this locus but has been

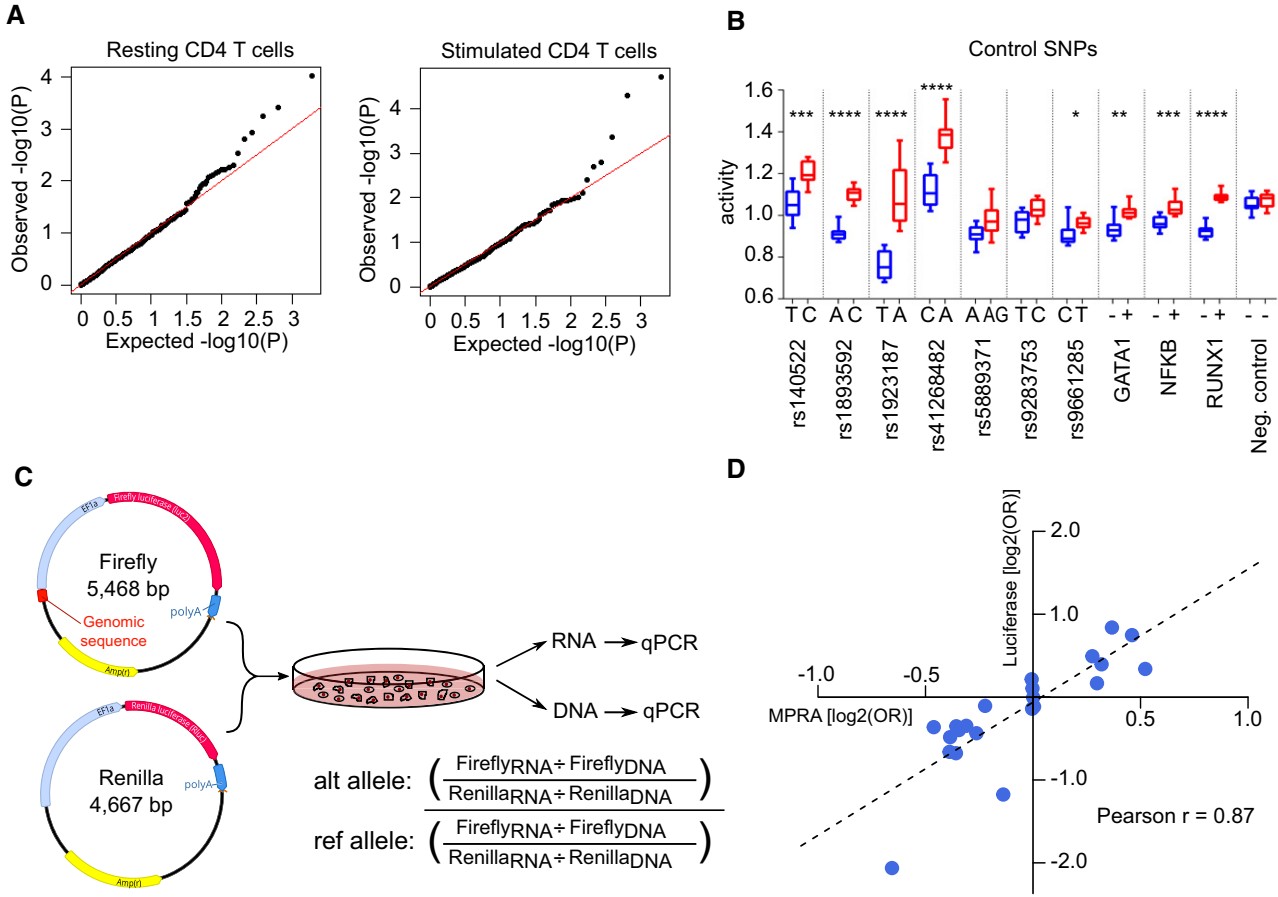

**Figure 2. Allele-specific expression-modulating effects in CD4 T cells.**

A  qq plots of the observed $-\log_{10}(P)$ values versus the expected $-\log_{10}(P)$ values under the null hypothesis for representative resting and stimulated CD4 T-cell samples.

B  Activity of each allele at 10 positive control SNPs and 1 negative control SNP in stimulated T cells. GATA1, NF-κB and RUNX1 constructs were designed to include a binding site for the indicated transcription factors (+) or with that site disrupted (−). Fixed-effects meta-analysis *P* value is shown: * < 0.05; ** < 0.01; *** < 0.001; **** < 0.0001. Box and whisker plots represent median and IQR (box) and min to max (whiskers). Exact *P* values are shown in Appendix Table S4.

C  Experimental workflow for validation experiment using a different promoter (EF1α), reporter gene (luciferase) and quantification method (qPCR).

D  Expression-modulating effect of each SNP [$\log_2(OR)$] as measured in MPRA and validation experiments. OR were calculated using the median activity of allelic constructs and are presented with respect to the risk allele (luciferase: *n* = 5, MPRA: *n* = 12).

hampered by strong LD (Farh *et al*, 2015; Huang *et al*, 2017; Fig 4B). In the MPRA, the same SNP (rs6927172) showed the strongest expression-modulating effect in both resting and stimulated T cells, with the risk allele consistently reducing transcription (Fig 4C, Appendix Figs S2 and S3). Further examination of this SNP revealed that it lies in a highly conserved region (Appendix Fig S4A) containing an experimentally validated NF-κB binding motif, to which all NF-κB dimers can bind (Wong *et al*, 2011). rs6927172 is located at position 10 within this 11-mer binding site, with the risk allele predicted to disrupt binding (Fig 4D, Appendix Fig S4B). To determine whether allele-specific NF-κB binding might account for the MPRA result, we transfected the MPRA plasmid library into CD4 T cells, immunoprecipitated NF-κB and quantified the plasmids to which it was bound. We confirmed that NF-κB differentially bound to rs6927172-containing plasmids in a manner consistent with the MPRA result and *in silico* prediction (Appendix Fig S4C). We next

investigated whether allele-specific NF-κB binding might also occur at the native locus in primary CD4 T cells. To do this, we isolated CD4 T cells from healthy donors who were heterozygous at rs6927172 and immunoprecipitated NF-κB to quantify the relative binding to each allele (Materials and Methods). We observed that in stimulated T cells, NF-κB exhibited reduced binding to the rs6927172 risk allele—consistent with the MPRA result (Fig 4E). Conversely, we did not detect allele-specific binding in resting T cells, which may reflect insufficient NF-κB signalling and suggests that the MPRA result in resting cells could be partly due to the transient activation that can occur following nucleofection (Zhang *et al*, 2014).

To determine whether differential NF-κB binding might affect enhancer strength, we exploited the fact that active enhancers are transcribed, producing enhancer-(e)RNAs whose abundance generally correlates with enhancer activity (Wu *et al*, 2014). Using an

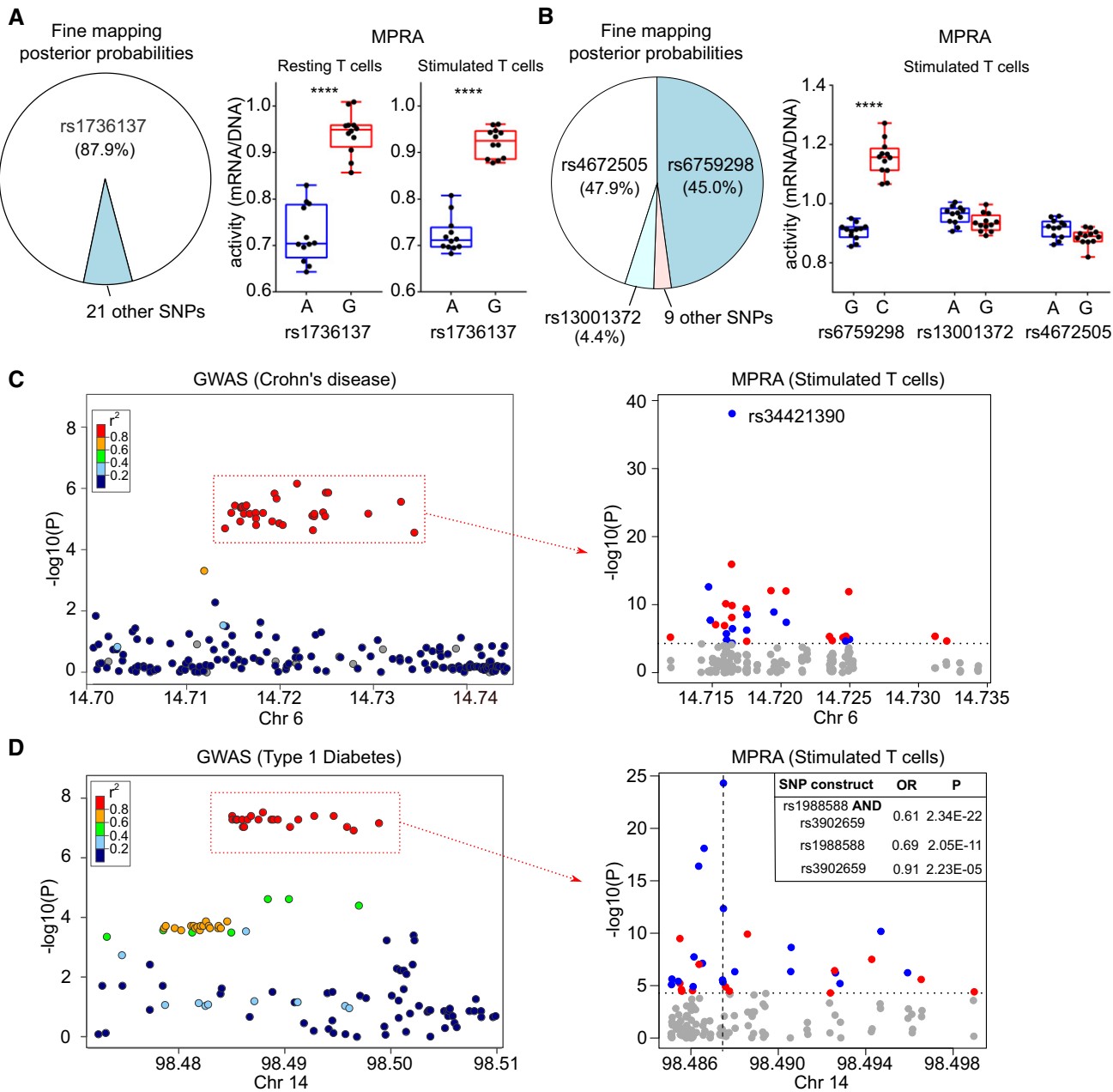

**Figure 3. MPRA in CD4 T cells identifies biological effects of disease associations.**

A  Pie chart depicting fine-mapping results (Huang *et al*, 2017; posterior probabilities) for an IBD-associated locus on 21q21 (left panel). MPRA results in resting (centre panel) and stimulated CD4 T cells (right panel) showing that the putative causal variant has significant expression-modulating effect.

B  Pie chart depicting Bayesian fine-mapping results for an AS-associated locus on 2p15 (left panel). MPRA results in stimulated T cells (right panel) showing that rs6759298 has a significant expression-modulating effect (the strongest of any variant at this locus) while the other candidate SNPs have negligible effects.

C  GWAS results (Liu *et al*, 2015) at a Crohn's disease and multiple sclerosis-associated locus on 6p23 (left panel). MPRA for candidate SNPs in stimulated T cells (right panel) identifying a SNP (rs34421390) with by far the greatest expression-modulating effect at this locus (blue, risk allele reduces expression; red, risk allele increases expression). Dotted horizontal line represents significance threshold (corrected for multiple testing).

D  GWAS results at a type 1 diabetes-associated locus on 14q32 (Onengut-Gumuscu *et al*, 2015; left panel). MPRA for the candidate SNPs in stimulated T cells (right panel). The construct with the largest and most significant effect contains the risk alleles for 2 SNPs (rs1988588 and rs3902659), each of which has a smaller effect when tested individually (position indicated by vertical dotted line). Dotted horizontal line represents significance threshold (corrected for multiple testing).

Data information: Box and whisker plots represent median and interquartile range (box) and min to max (whiskers). ****FDR-corrected meta-analysis *P* < 0.0001. Exact *P* values are shown in Appendix Table S4.

allele-specific expression assay, we compared the amount of eRNA transcribed from each allele in stimulated heterozygous CD4 T cells—thus ensuring that external factors would affect both alleles equally. We found that transcription of the eRNA was lower from the risk allele, in which the NF-κB binding site is disrupted (Fig 4F). This suggests that the disease-associated haplotype diminishes enhancer activity by reducing NF-κB binding—potentially linking the genetic association to a specific functional deficit.

During inflammatory responses, NF-κB binding has been reported to direct dynamic super-enhancer formation (Brown *et al*, 2014). To better define the functional consequences of allele-specific NF-κB binding at this locus, we performed H3K27 acetylation (H3K27ac) ChIP-sequencing in stimulated CD4 T cells from major and minor allele homozygotes at rs6927172. This facilitated a genome-wide comparison of active regulatory regions and enabled us to characterise the effect of rs6927172 on enhancer activity. We observed consistently stronger enhancer activity at this locus in major allele homozygotes compared with minor (risk) allele homozygotes (Appendix Fig S5A). To improve peak-calling and generate representative data sets, we combined the genotypic replicates for subsequent analysis. Using the Rank Ordering of Super-Enhancers (ROSE) algorithm (Whyte *et al*, 2013), we found that rs6927172 was located within a 45.5 kb super-enhancer in major allele homozygotes (Fig 4G, Appendix Fig S5A). This super-enhancer appears to be T cell-specific and potentially activation-specific, since it is also present in stimulated Th17 cells, but not in 27 other primary tissues nor in five other immune cell types (Hnisz *et al*, 2013). Consistent with this, we found that many of the transcription factors predicted to bind within the constituent elements of the super-enhancer were involved in T-cell activation (Appendix Fig S5B and C). In contrast to the strong enhancer activity in major allele homozygotes, there was negligible enhancer activity at the rs6927172 locus in minor allele homozygotes (Fig 4G, Appendix Fig S5A). Indeed, while enhancer activity was detected 1.5 kb upstream and 18.8 kb downstream of this SNP (extending to the 5′ and 3′ ends of the annotated super-enhancer), the overall enhancer strength across this region was fourfold lower in the presence of the risk allele, and super-enhancer formation was accordingly disrupted (Fig 4G, Appendix Fig S5A).

To understand why disrupting the formation of an NF-κB-driven super-enhancer might predispose to multiple immune-mediated diseases, we next investigated the genes that it regulated. Using available promoter capture Hi-C data from stimulated CD4 T cells, we confirmed that the majority of super-enhancer interactions were either with the promoter of *TNFAIP3* or with a region ∼ 41 kb downstream of *TNFAIP3* that also interacts with the *TNFAIP3* promoter (Fig 4H). Consistent with this, we found that rs6927172 genotype correlated with expression of *TNFAIP3*—but not other genes at this locus—in CD4 T cells from patients with active IBD (Fig 4I). Expression of *IL20RA*, which was suggested to be causal based on experiments in cell lines (McGovern *et al*, 2016; Wu *et al*, 2019), could not be detected in primary CD4 T cells and is not expressed in primary immune cells according to publicly available data sets (Fernandez *et al*, 2016; Schmiedel *et al*, 2018; Appendix Fig S5D). In contrast, *TNFAIP3* is highly expressed in effector CD4 T cells (Schmiedel *et al*, 2018; Appendix Fig S5E) and encodes A20, a key negative regulator of NF-κB signalling and an early target gene of NF-κB (Lee *et al*, 2000).

Collectively, these data are consistent with a model in which NF-κB signalling in stimulated CD4 T cells leads to the formation of a super-enhancer that upregulates a key NF-κB inhibitor—thereby limiting inflammatory responses. This regulatory circuit can be disrupted by a common expression-modulating variant, such that NF-κB binding and enhancer activity are diminished in the presence of the risk allele. This would be predicted to lead to excessive inflammatory responses in CD4 T cells, consistent with the pleiotropic association with immune-mediated disease.

### The NF-κB binding site disrupted by rs6927172 regulates *TNFAIP3* expression and inflammatory responses in CD4 T cells

To test whether our proposed model was correct, we sought to delete the NF-κB binding site in primary CD4 T cells using CRISPR-Cas9. Efficient genome editing in primary T cells usually requires the cells to be pre-activated (Hendel *et al*, 2015), but a method was recently described for editing resting T cells (Seki & Rutz, 2018). Since we wished to study the effects of editing upon subsequent T-cell activation, we similarly optimised conditions for editing resting T cells (Materials and Methods, Appendix Fig S6A and B). To reduce the chance that an observed effect might be due to off-target activity, we designed several gRNAs that flanked the rs6927172-containing NF-κB binding site and used these in different combinations (Fig 5A). We achieved mean editing rates of 60–70% for three of the four gRNA combinations, of which ∼ 80% of predicted indels ablated the NF-κB binding site (Fig 5B). Of note, the lower editing rate observed with the fourth gRNA combination probably reflects steric hindrance between Cas9-gRNA ribonucleoproteins (RNPs) since the offset between gRNAs was only 4 bp (Ran *et al*, 2013).

We next investigated how deleting the NF-κB binding site would affect transcription locally. After RNP nucleofection, CD4 T cells were rested for 48 h for editing to occur and then stimulated for 24 h (Fig 5C, Materials and Methods). To specifically quantify RNA that was transcribed during T-cell activation—and after editing—we added 5-ethynyl uridine (EU) at the time of stimulation to facilitate nascent RNA capture (Materials and Methods). RNA that incorporated this modified base was purified, and expression of protein-coding genes within 1.5 Mb of the deletion site was measured and normalised to a non-targeting control. Of the six genes tested, only *TNFAIP3* expression was significantly altered (Fig 5D). Moreover, individual deletions of the other candidate SNPs within the disease-associated super-enhancer did not significantly alter *TNFAIP3* expression—consistent with dysregulation of enhancer activity being specific to rs6927172 (Appendix Fig S6C).

To understand the biological consequences of this effect, we next examined markers of T-cell activation. Using a fluorescently tagged gRNA that is detectable by flow cytometry, we distinguished CD4 T cells that contained RNPs (and were likely to have been edited) from those that did not. Analysing these populations separately, we observed a specific increase in CD69 expression, an early marker of T-cell activation, in the RNP-containing cells, that was present not in the non-targeting control, nor in RNP-negative cells from the same transfection (Fig 5E). This indicated that deletion of the NF-κB binding site, which is physiologically disrupted by rs6927172, leads to increased T-cell activation.

To further explore the underlying mechanism, we used flow cytometry to quantify IκBα phosphorylation, a key step in NF-κB

signalling (Zandi *et al*, 1997). After normalising to the non-targeting control, the increase in phospho-IκBα was found to directly correlate with the overall editing efficiency (Fig 5F)—suggesting that NF-κB signalling increases proportionally with deletion of the NF-κB binding site. To understand how this would affect CD4 T-cell effector function, we quantified cytokine production and found that deletion of the NF-κB binding site increased the production of effector cytokines from all major T helper cell

lineages, consistent with unrestrained inflammatory responses (Fig 5G). Finally, to confirm that these results were indicative of a *TNFAIP3*-dependent effect, we directly disrupted *TNFAIP3* using RNPs and showed that this phenocopied the observed effects, with marked increases in T-cell activation (Appendix Fig S6D) and effector cytokine production (Appendix Fig S6E), consistent with the known role of A20 in regulating inflammation (Lee *et al*, 2000).

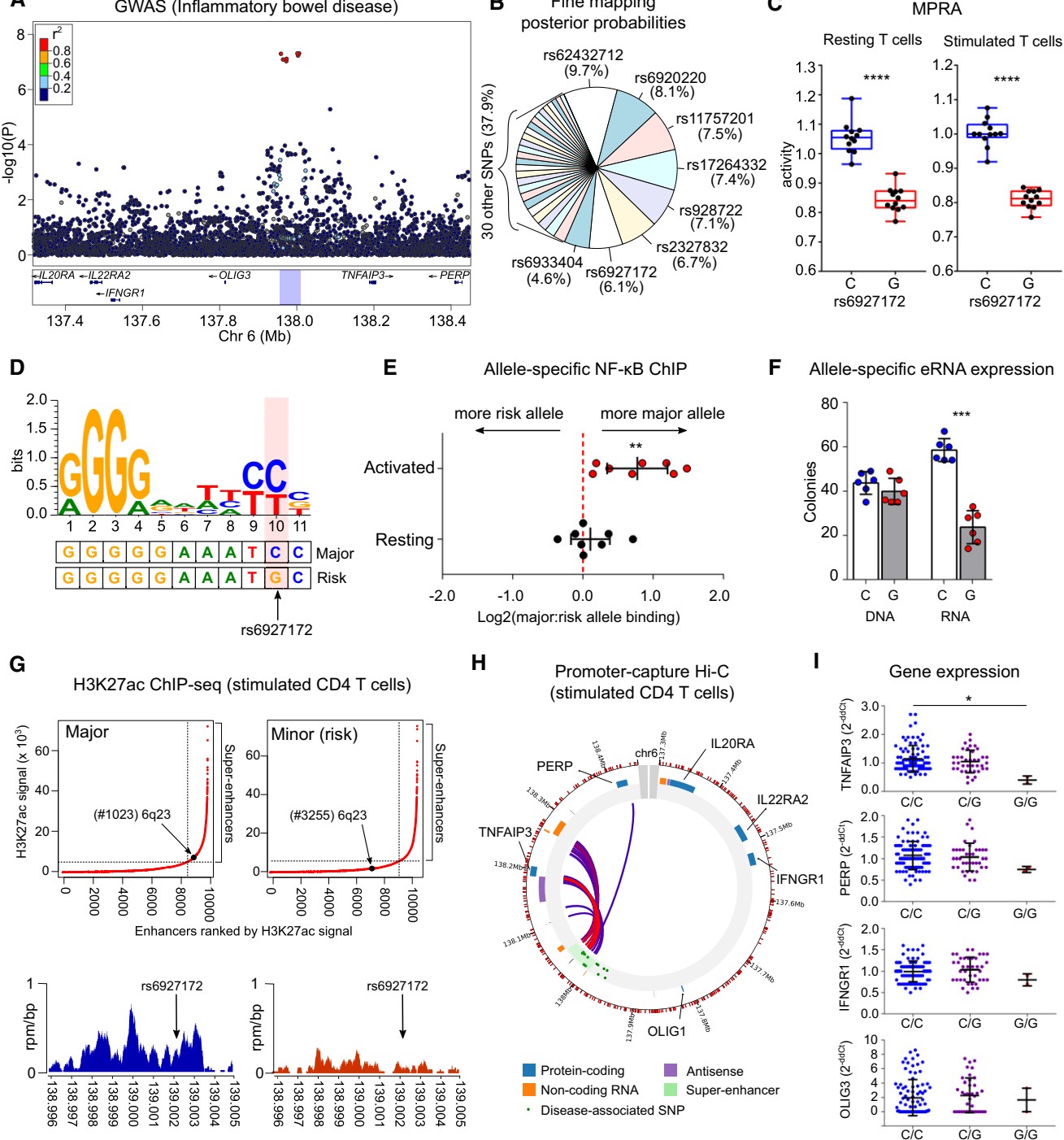

**Figure 4.**

**Figure 4. MPRA in CD4 T cells identifies an expression-modulating variant that disrupts NF-κB binding and enhancer function.**

A IBD GWAS results (Liu *et al*, 2015) at a multi-disease-associated locus on chromosome 6q23.

B Fine-mapping results (Huang *et al*, 2017; posterior probabilities) for candidate SNPs at this locus.

C A single variant (rs6927172) has the largest and most significant expression-modulating activity in resting (left panel) and stimulated CD4 T cells (right panel) with the risk allele reducing transcription. Plots represent median and IQR (box) and min to max (whiskers). FDR-corrected meta-analysis *P* value shown.

D Sequence logo for an experimentally validated NF-κB binding motif (Wong *et al*, 2011). The genomic sequence around rs6927172 is aligned below.

E Allele-specific NF-κB binding in CD4 T cells from rs6927172 heterozygotes, demonstrating reduced NF-κB binding to the risk allele following stimulation (*n* = 8; one-sample *t*-test, two-tailed).

F Allele-specific expression of enhancer RNA in heterozygous CD4 T cells. DNA used for technical control (*n* = 6; paired *t*-test; two-tailed).

G Genome-wide H3K27ac ChIP-seq in stimulated CD4 T cells from major and minor allele homozygotes at rs6927172 (*n* = 6). Upper panels show input-normalised H3K27ac signals plotted against enhancer rank. Super-enhancers are defined above the inflection point of the curve. Lower panels show H3K27ac reads from a major (left) and a minor (risk) allele homozygote (right) in a 9 kb window around rs6927172.

H Promoter capture Hi-C plot depicting interactions of the 6q23 super-enhancer in stimulated CD4 T cells.

I Expression of genes on 6q23 in CD4 T cells from 131 patients with active IBD, stratified by rs6927172 genotype (qPCR; one-way ANOVA). Error bars represent SD. Expression of *IL20RA* and *IL22RAR2* not detected.

Data information: Data represent mean ± SEM, unless indicated. *P < 0.05; **P < 0.01, ***P < 0.001, ****P < 0.0001. Exact P values are shown in Appendix Table S4.

Collectively, these data identify an NF-κB-driven regulatory circuit which constrains T-cell activation through the dynamic formation of a super-enhancer that drives expression of *TNFAIP3*, a key NF-κB inhibitor. In primary CD4 T cells, this circuit is disrupted—and super-enhancer formation prevented—by the risk variant at rs6927172, thus revealing the molecular and cellular consequences of a pleiotropic disease association.

## Discussion

A fundamental goal of GWAS was to better understand disease biology (Visscher *et al*, 2017). As such, despite widespread success in variant discovery, this goal remains unfulfilled—since we have not yet transitioned from lists of associated SNPs to insights into disease mechanisms. Here, we have adapted an MPRA to simultaneously assess the functional effects of hundreds of non-coding genetic variants in primary CD4 T cells—the cell type whose regulatory DNA is most enriched for immune-mediated disease SNPs (Farh *et al*, 2015). By analysing each SNP individually, this method bypasses the limitations of LD and enables putative causal variants to be identified via their biological effects. Unlike fine-mapping, this approach does not attempt to refine GWAS statistics and so does not provide a specific estimate of causality for each SNP. However, by identifying SNPs that causally change biology, this method can resolve the functional consequences of disease-associated genetic variation and provide a focus for studying disease mechanisms. Importantly, this approach is broadly applicable and could be used to identify putative causal variants at any disease-associated locus that overlaps with T-cell regulatory elements—even those that cannot be fine-mapped—thus overcoming a major bottleneck in the transition from genetic variants to disease mechanisms.

To illustrate the value of this approach, we used the MPRA results as a basis for investigating disease mechanisms at a pleiotropic locus that has been linked to six different immune-mediated diseases but cannot be fine-mapped. In doing so, we uncovered a regulatory circuit that constrains T-cell activation through the dynamic formation of *TNFAIP3* super-enhancer and show how this can be disrupted by a common, expression-modulating variant that perturbs NF-κB binding—consistent with the known vulnerability of super-enhancers to perturbation of their components

(Hnisz *et al*, 2013). Altogether, this identifies a biological mechanism that is likely to be broadly involved in human autoimmune disease. Indeed, while exuberant effector CD4 T-cell responses have been implicated in the initiation and perpetuation of all of the diseases linked to rs6927172 (Bluestone *et al*, 2015), evidence of how common genetic variation might contribute to this has previously been lacking.

A key strength of performing MPRA, and follow-up experiments, in primary cells is that this provides greater confidence that the results are physiologically relevant. For example, these data show that in primary CD4 T cells, the biological effect of this multi-disease-associated haplotype is solely mediated by *TNFAIP3*, and not by any of the other genes at the locus. Several lines of evidence support T cells as the relevant cell type for this association, including the presence of a T cell-specific super-enhancer involving the lead SNP (Hnisz *et al*, 2013) and the key role that T cells are known to play in all the diseases linked to rs6927172 (Bluestone *et al*, 2015). Moreover, a very recent study of chromatin accessibility identified an allele-specific effect of rs6927172 in stimulated CD4 T cells that was not detected in other immune cell types (Calderon *et al*, 2019). By resolving the downstream consequences of this effect—both on local transcription and, in turn, on T-cell responses—we extend this observation and identify a mechanism consistent with a broad predisposition to immune-mediated disease.

Of note, a similar mechanism was previously reported for a different locus that is located downstream of *TNFAIP3* and associated with SLE, but not with any the other diseases linked to rs6927172 (Adrianto *et al*, 2011). At this low frequency haplotype, which is not in LD with rs6927172 ($r^2$ = 0.001), the putative causal variant also alters NF-κB binding and interacts with the *TNFAIP3* promoter (Adrianto *et al*, 2011). That two distinct disease-associated loci have similar functional consequences highlights the importance of *TNFAIP3* in human autoimmunity and may point to cell type-specific effects. For example, the SLE-only locus does not interact with *TNFAIP3* in stimulated CD4 T cells (Javierre *et al*, 2016) but has strong enhancer activity in transformed B cells (Whyte *et al*, 2013)—a cell type linked to SLE pathogenesis and enriched for SLE-associated variants (Farh *et al*, 2015).

This adapted MPRA is subject to the same limitations as regular MPRA; in that, each construct is tested in a plasmid, out of its native genomic context—reinforcing the importance of studying regions in

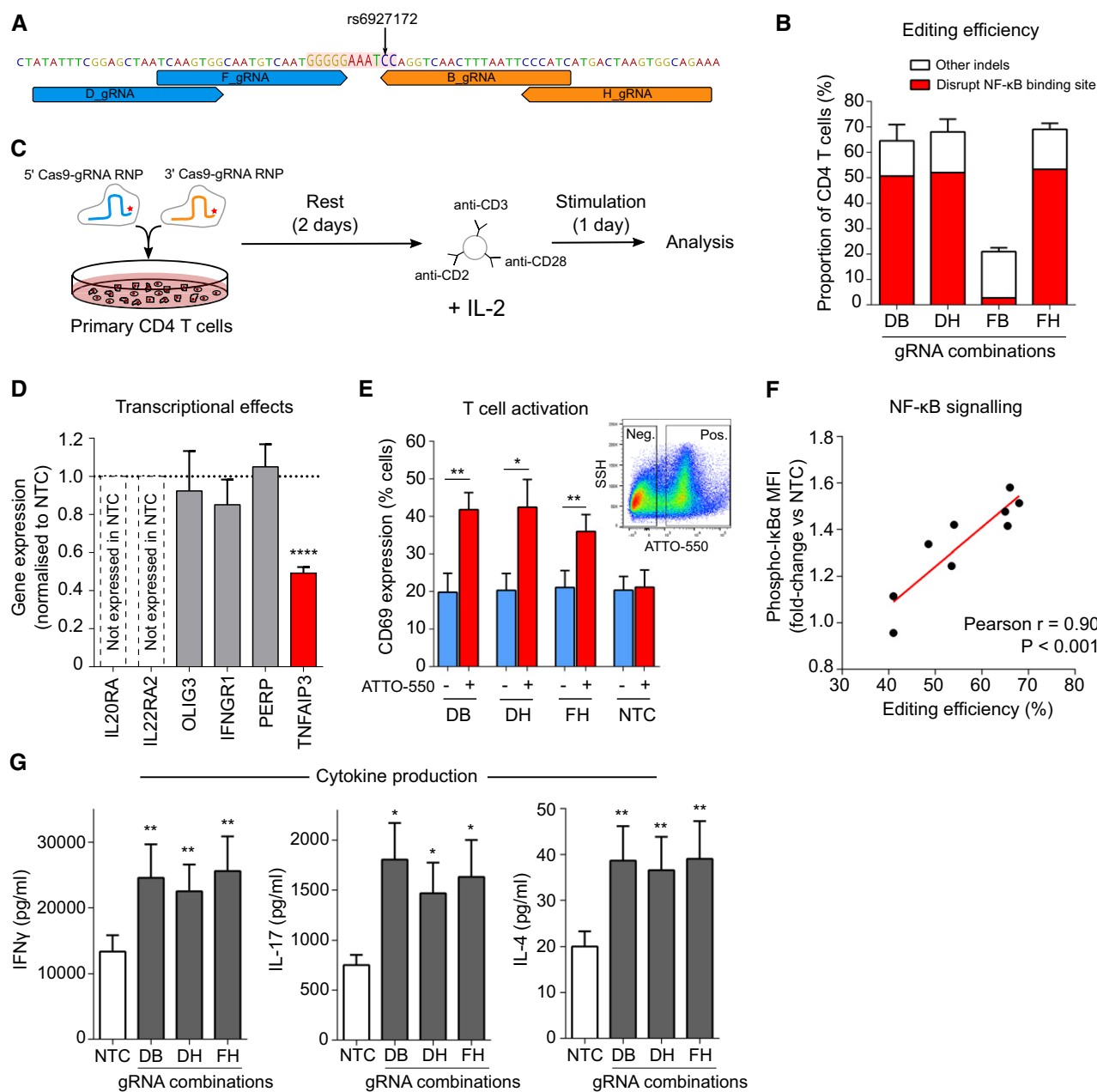

**Figure 5. Deletion of the NF-κB binding site, disrupted by rs6927172, dysregulates *TNFAIP3* expression and increases CD4 T cell activation.**

A   Location of gRNAs flanking the NF-κB binding site (highlighted).

B   Primary CD4 T-cell editing efficiency for indicated combinations of 5′ and 3′ gRNAs (*n* = 6 for DB, DH and FH and *n* = 2 for FB). Distribution of indels assessed using ICE.

C   Experimental workflow: equimolar amounts of 5′ and 3′ gRNA-containing RNPs (fluorescently tagged with ATTO-550) were nucleofected into resting CD4 T cells, which were stimulated after 48 h with anti-CD2/3/28 microbeads and IL-2.

D   Expression of genes on 6q23 in EU-containing mRNA (EU added at time of stimulation) showing that deletion of the NF-κB binding site specifically reduces transcription of *TNFAIP3*, but not other genes at this locus (*n* = 6; one-sample *t*-test). Representative data from the DH gRNA combination.

E   Expression of CD69, an activation marker, following CRISPR editing with indicated gRNA combinations or the non-targeting (negative) control (NTC)—data shown for ATTO-550 positive (RNP-containing) and negative cells (*n* = 6; paired *t*-test, one-tailed). Inset flow cytometry plot depicting representative gating of ATTO-550-positive and ATTO-550-negative cells.

F   Correlation between editing efficiency (total indel rate) and levels of phosphorylated IκBα in CD4 T cells (normalised to the mean fluorescence intensity in NTC; *n* = 9, linear regression).

G   Secretion of effector cytokines following deletion of the NF-κB binding site—Th1 (left panel, IFNγ), Th17 (centre panel, IL-17A) and Th2 subsets (right panel, IL-4; *n* = 6, paired *t*-test, one-tailed).

Data information: Data represent mean ± SEM. **P* < 0.05; ***P* < 0.01; *****P* < 0.0001. Exact *P* values are shown in Appendix Table S4.

relevant cells. Extending MPRA to other primary cell types, particularly to study disease-specific loci, is a logical next step. These data also highlight the value of considering MPRA results as hypotheses to be experimentally tested, rather than as definitive insights into isolation. Indeed, we show that using multiple biological replicates adds considerable power to identify expression-modulating effects, and so characterising the functional consequences of any result is essential.

In summary, we have developed a scalable method that can distil disease-associated haplotypes down to specific functional variants in relevant primary cells, thereby generating testable hypotheses into disease mechanisms—even within gene deserts—while overcoming some of the limitations of statistical fine-mapping. This can provide important insights into disease biology and represents a framework by which much of the potential of GWAS in immune-mediated disease could finally be realised.

# Materials and Methods

## Locus selection and library design

Fourteen loci were selected from immunochip studies in 10 immune-mediated diseases. Inclusion criteria: no coding or well-characterised non-coding genes in the extended haplotype (tagged by SNPs with $r^2 > 0.8$ with the lead variant and extended by 100 bp each side). Oligonucleotides were designed to test every candidate SNP and tile each locus at 50 bp intervals. Sliding windows were used for allelic constructs, so that 1/3, 1/2 and 2/3 of the construct were 3′ of the variant, as described previously (Ulirsch et al, 2016). If adjacent SNPs were within 114 bp, extra oligonucleotides were synthesised to test combinations of risk alleles. Allelic constructs were tagged by 30 unique 11nt barcodes and tiling constructs by six unique barcodes. Ten positive control SNPs were included: five expression-modulating variants (Tewhey et al, 2016), two single variant eQTLs (Farh et al, 2015) and three synthetic SNPs (designed to include/disrupt a known binding motif: GATA1/3 motif = TGATAG; RUNX1 motif = TGTGGTTT; NF-κB motif = GGGGGAATCCC). For the tiling analysis, 2 kb control regions were used (positive = T-cell super-enhancers: chr21:36421330-36423329, chr1:198626200-198628199; negative = gene deserts with no enhancer activity: chr4:29562525-29564524, chr4:34780413-34782412, hg19). In total, 99,990 170 bp oligonucleotides were synthesised (Twist Biosciences) to contain the 16nt universal primer ACTGGCCGCTTCACTG, 114nt genomic sequence, KpnI | XbaI restriction sites, an 11nt barcode and the 17nt universal primer AGATCGGAAGAGCGTCG.

## Oligonucleotide library cloning

Oligonucleotides were amplified by emulsion PCR (Micellula DNA Emulsion & Purification Kit, Chimerx) using primers containing SfiI restriction sites. 200 ng of the amplified library was digested with SfiI (NEB) and cloned into SfiI-digested pGL4.10M vector. Resulting plasmids were purified (Plasmid Plus Maxi kits, Qiagen), quantified (Nanodrop 1000, Thermo Fisher) and sequenced to check barcode representation. 2 μg purified plasmids were digested (KpnI/XbaI, NEB) and ligated with a KpnI/XbaI-digested fragment containing a promoter and EGFP (promoters: minimal promoter, SV40 (from CBFRE-EGFP), RSV (from pRSCgfp-hAIM2) and EF1α (from pOTTC407-pAAV EF1a eGFP)). Ligation products were transformed into Escherichia coli, purified and quantified, as described. The resulting MPRA plasmid library was sequenced (MiSeq) to confirm barcode representation.

## CD4 T-cell purification, transfection and cell culture

Source Leukocytes, purified from healthy donors, were obtained from Massachusetts General Hospital (MGH) Blood Transfusion Service. Peripheral blood mononuclear cells were isolated by density centrifugation (Histopaque 1077, Sigma) and CD4 T cells positively selected, as described previously (Lee et al, 2011). Purity was confirmed by flow cytometry (> 95%). CD4 T cells were split 2:1 for immediate nucleofection (resting) or stimulation. Stimulation was performed for 4 days using recombinant human IL-2 (10 ng/ml, PeproTech) and Anti-Biotin MACSiBead Particles loaded with CD2/3/28 antibodies (bead-to-cell ratio 1:2, Miltenyi). A minimum of six technical replicates (nucleofections) were performed per sample. Each replicate: 5 μg vector, 5 M CD4 T cells, 100 μl "1 M" nucleofection solution (Chicaybam et al, 2013), Nucleofector 2b program V024 (resting T cells) or T023 (stimulated T cells). After nucleofection, 500 μl pre-warmed media was added to the cuvette and cells were transferred to a 6-well flat-bottomed plate (final volume 5 ml/well) and cultured at 37°C, 5% $CO_2$. Cell culture media: IMDM (Thermo Fisher), 20% FBS (Thermo Fisher), 1% non-essential amino acids (Thermo Fisher), 2 mM GlutaMAX (Thermo Fisher) and 1% sodium pyruvate (Thermo Fisher). No antibiotics were used. 24 h after nucleofection, cells were harvested, pooled and lysed in RLT Plus buffer (Qiagen).

## Jurkat culture and transfection

Jurkats (Clone E6-1) were cultured in RPMI-1640, 10% FBS (Thermo Fisher), 1% non-essential amino acids (Thermo Fisher), 2 mM GlutaMAX (Thermo Fisher) and 1% sodium pyruvate (Thermo Fisher). No antibiotics were used. Transfections were performed at rest or following stimulation (conditions as for CD4 T cells). Nucleofection conditions: 2 μg vector, 1 M Jurkats, 100 μl Amaxa Cell Line Nucleofector V solution and program X001. Cells were harvested, pooled and lysed after 24 h.

## Flow cytometry

CD4 T-cell purity, composition and transfection efficiency were assessed by flow cytometry (BD LSR II). Purity and composition panel (stained in 250 μl): CD4 APC (#357408), CCR4 BV421 (#359414), CCR6 AF700 (#353434), CD3 FITC (#300306), CD62L PerCP/Cy5.5 (#304824), CXCR3 PE/Dazzle 594 (#353736) and CD45RA PE/Cy7 (#304126)—all used at 1:100 dilution (all BioLegend), Zombie Aqua Fixable Viability Kit (BioLegend) and Fc receptor blocking reagent (Miltenyi). Transfection efficiency panel: EGFP, CD4 APC (as above), Zombie Aqua Fixable Viability Kit, Fc receptor blocking reagent. Data were gated using FlowJo v10 (BD).

## Library preparation

Lysates were DNA depleted (gDNA eliminator column, Qiagen), and RNA was extracted using a RNeasy Plus micro kit (Qiagen). For

library preparation, 1 μg RNA was treated with TURBO DNAse (Thermo Fisher) and reverse transcribed (SuperScript IV VILO, Thermo Fisher). DNA removal was confirmed by qPCR. Sequencing libraries were prepared by PCR (30 cycles) using PfuUltra II polymerase (Agilent) and custom primers that annealed to a 3′ site within EGFP (F) and the 3′ universal primer site in the oligonucleotide (R). Amplified libraries were cleaned using AMPure XP beads (Beckman Coulter): 0.6×, 1.6× and 1.0×. Four sequencing libraries were made from the input MPRA plasmid library (50 ng vector, 18 PCR cycles). Libraries were sequenced in pools of 6 (Illumina HiSeq2500, 50 bp, single-end reads).

### Primers

Primer sequences are provided in Appendix Table S2.

### MPRA analysis

#### Pre-processing

Barcode counts were obtained from FASTQ files after quality control (FastQC). To be counted, a sequenced barcode had to match an oligonucleotide barcode and be followed by $\geq 10$ bases of the expected constant sequence. To be a successful transfection, $\geq 70\%$ of the barcode library had to be represented. Raw count data were normalised (CPM) and filtered to remove barcodes with median CPM < 0.5 in RNA or DNA samples. Barcode counts for identical constructs were then collapsed (summed) and quantile normalised.

#### Principal component analysis

Principal component analysis (PCA) was performed on pre-processed data from RNA samples using the *prcomp* function in R. The data were centred and scaled to have unit variance, and then, singular value decomposition was performed. No components were omitted.

#### Pairwise correlation analysis

The transcriptional activity of each construct was calculated by dividing the normalised construct-level count (mRNA) by the median count from the same construct in the input library (DNA). Correlation matrices were created using the *cor* function (Pearson correlation) in R.

#### Tiling analysis

To measure enhancer activity, we used the *sharpr2* package (Wang *et al*, 2018) in R. This was used because (i) the tiled regions were of different sizes, (ii) the offset between constructs (50 bp) was not a factor of the length of genomic sequence (114 bp), and (iii) we could include reference allele SNP constructs to improve genomic coverage. After removing alternate allele constructs, the median counts for the remaining constructs in RNA and DNA were analysed. The regulatory scores for each region were based on standardised log(RNA/PLASMID), and a regional FWER cut-off (0.05) was used to identify high-resolution driver elements indicative of enhancer activity.

#### SNP analysis

For the SNP analysis, we used QuASAR-MPRA (Kalita *et al*, 2018) in the *QuASAR* package in R. After removing enhancer constructs,

964/970 SNP constructs were available for analysis. As this can only analyse one sample at a time, a standard fixed-effect meta-analysis was used to combine the results for biological replicates. To do this, we used the logit transformation of the proportion of reference reads in RNA ($\beta_l$) and the standard error of this estimate ($\sigma_{\beta_l}$). We then calculated logit transformation of the proportion of reference reads in DNA ($\beta_0$). So, for $k$ samples:

$$\beta^*_{l.\text{adj}} = \frac{1}{w^*_l} \sum_i^k (\hat{\beta}_{i,l} - \beta_{0,l}) w_{i,l}$$

where $w_{i,l} = \frac{1}{\sigma^2_{\beta_{i,l}}}$ and $w^*_l = \sum_i^k w_{i,l}$. We then calculated a meta-analysis $Z$ score and $P$ value:

$$Z = \frac{\beta^*_{l.\text{adj}}}{\hat{\sigma}^*_l}$$

where $\hat{\sigma}^*_l = \frac{1}{\sqrt{w^*_l}}$.

FDR correction for multiple testing was applied (Benjamini & Hochberg, 1995).

### Luciferase-based validation

The allelic constructs for the lead MPRA SNP at each locus were synthesised as geneblocks with flanking restriction sites (5′ KpnI, 3′ BamHI; IDT). For one haplotype (14q32), the sequence was too GC rich for synthesis and so was PCR amplified from a major allele homozygote and site-directed mutagenesis used to make the alternate allele construct (Q5 Site-Directed Mutagenesis Kit, NEB). Additional geneblocks for two positive controls and five SNPs with no expression-modulating activity were synthesised (IDT). Geneblocks were digested (KpnI/BamHI) and ligated into a custom Firefly luciferase vector (VectorBuilder) to lie proximal to the luciferase promoter (EF1α). The ligation product was transformed into *E. coli*, sequenced to confirm geneblock insertion (Genewiz) and purified. For each geneblock, equimolar amounts of the Firefly vector and a custom Renilla luciferase vector (total 5 μg vector) were nucleofected into resting CD4 T cells. DNA and RNA were extracted after 24 h. 200 ng RNA was DNAse treated (TURBO DNase, Thermo Fisher) and reverse transcribed (SuperScript IV VILO, Thermo Fisher). Quantification of Firefly and Luciferase genes was performed in triplicate using qPCR. The results were normalised using an adapted Delta-Delta-Ct method in which the Cts for Firefly and Renilla (mRNA) were first normalised to their respective DNA Cts (to control for any imbalance in the input mix) and then each Firefly Delta-Ct was normalised to the Renilla Delta-Ct (to control for transfection efficiency)—producing a measure of expression-modulating activity. The activities of reference and alternate alleles were compared to determine the SNP effect. Five biological replicates were performed.

### Fine-mapping ankylosing spondylitis (AS) association on 2p15

AS summary statistics (International Genetics of Ankylosing Spondylitis C *et al*, 2013) were obtained from the GWAS catalog, and SNPs in chr2:62518445..62618445 (hg19) were extracted. Approximate Bayes factors summarising the association at each SNP, and thus the posterior probabilities for each SNP being causal,

were calculated (Wakefield, 2009)—assuming a single causal variant model. The 99% credible set contained four SNPs. Recent work has shown that this method can be biased (Hutchinson *et al*, 2020) and so we used the *corrcoverage* R package to correct any bias, identifying a 99% credible set containing three SNPs: rs6759298, rs4672505 and rs13001372 (corrected coverage 99.2%).

### NF-κB binding site analysis

A common NF-κB motif was identified from protein-binding microarray data (Additional File 2 from Wong *et al*, 2011). In brief, the z-scores for the affinity of nine NF-κB dimers for each 11-mer sequence were combined using Stouffer's method to calculate the statistical significance of the overall binding. After Bonferroni correction, 100 statistically significant 11-mer sequences were identified ($P_{adjust} < 0.05$) which had positive z-scores for every dimer. These sequences were used to generate a common NF-κB motif logo using WebLogo (Crooks *et al*, 2004).

### NF-κB immunoprecipitation following MPRA library nucleofection

CD4 T cells were purified and nucleofected with the MPRA plasmid library, as described ($n = 4$). After 24 h, cells were harvested, resuspended in fresh media ($10^6$ cells/ml) and cross-linked (37% Formaldehyde to final concentration 1% for 10 min). This was then quenched with glycine (final concentration 0.125 M) for 5 min. After washing, cell pellets lysed for 10 min in lysis buffer with Complete Mini EDTA-free Protease Inhibitor (Roche). Lysis buffer: 50 mM HEPES pH 7.9, 140 mM NaCl, 1 mM EDTA pH 8.0, 10% v/v Glycerol, 0.5% v/v IGEPAL CA-630, 0.25% v/v Triton X-100. Two cycles of sonication were performed (Bioruptor Pico, Diagenode) to remove contaminants without chromatin shearing. Triton X-100 and NaCl were added to final concentrations of 1% and 100 mM, respectively. 10 μg of sheared chromatin were cleared by centrifugation (21,000 *g*, 10 min, 4°C), and immunoprecipitation was performed overnight at 4°C using an anti-NFkBp65 antibody (1:100, clone D14E12; Cell Signaling) or an isotype control (1:500, rabbit IgG, Abcam; ab172730) with the SimpleChIP Plus kit (Cell Signaling). Sequencing libraries were prepared as described earlier (26 PCR cycles).

### Allele-specific NF-κB ChIP

A 100 ml blood sample was obtained from eight healthy individuals, heterozygous at rs6927172—identified via the NIHR BioResource. All participants provided written informed consent. Ethical approval was provided by the Cambridgeshire Regional Ethics Committee (REC:08/H0308/176). All experiments conformed to the principles set out in the WMA Declaration of Helsinki and the Department of Health and Human Services Belmont Report. CD4 T cells were purified and left resting or stimulated for 4 days, before harvesting, cross-linking, quenching and lysing, as described. After washing (wash buffer: 10 mM Tris-HCl pH 8.0, 200 mM NaCl, 1 mM EDTA pH 8.0, 0.5 mM EGTA pH 8.0), nuclei were resuspended in 200 μl shearing buffer/$10^7$ cells and sonicated for nine cycles (30 s ON/30 s OFF) using a Bioruptor Pico (shearing buffer: 0.1% w/v SDS, 1 mM EDTA, 10 mM Tris-HCl pH 8.0). Triton X-100 and NaCl were added as described earlier. NF-κB ChIP was performed as described.

To assess for allele-specific binding, the NF-κB-bound DNA was genotyped for rs6927172 in triplicate (TaqMan) alongside pre-mixed DNA from a minor and a major allele homozygote at rs6927172. A series of ratios of minor:major allele homozygote DNA were used (from 4:1 to 1:4) to create a standard curve, against which the ratios obtained in the NF-κB ChIP samples were compared.

### Allele-specific eRNA analysis

DNA and RNA were extracted from stimulated CD4 T-cell lysates (non-nucleofected). Genotyping was performed to identify 6 × rs6927172 heterozygotes. RNA was TURBO DNase-treated and reverse-transcribed as described earlier. Nested PCR was performed to amplify the region around rs6927172 from genomic DNA and cDNA. PCR amplicons were gel-purified (Zymoclean Gel DNA Recovery Kit, Zymo) and diluted to 8 ng/μl. 8 ng (5:1 insert:vector ratio) was ligated into a Zero Blunt TOPO vector (Thermo Fisher) and transformed into *E. coli*. For each sample, 96 colonies were genotyped to quantify allelic ratios.

### H3K27ac ChIP-seq and analysis

A 100 ml blood sample was obtained from three major and three minor allele homozygotes at rs6927172—identified via the NIHR BioResource. CD4 T cells were purified and stimulated for 4 days, before harvesting, cross-linking, quenching, lysing and nuclei shearing as described. 2% input samples were stored prior to immunoprecipitation, which was performed overnight at 4°C with an anti-H3K27ac antibody (1:250, Abcam; ab4729) or an isotype control (1:500, rabbit IgG, Abcam; ab172730) using the SimpleChIP Plus kit. 50 ng of immunoprecipitated DNA or input sample was used for library preparation (10 amplification cycles, iDeal Library Preparation kit, Diagenode). Libraries were sequenced in pools of 8, with each pool sequenced in two lanes of an Illumina HiSeq2500 high output flow-cell (50 bp, single-end reads): median 50.4 M reads (H3K27ac ChIP) and 79.6 M reads (input). Sequencing reads were trimmed using TrimGalore! (Phred score 24), filtered to remove reads < 36 bp and aligned to the human genome (hg19) using Burrows-Wheeler Aligner (BWA). Aligned reads were converted to BAM files, sorted and technical duplicates merged before indexing using SAMtools (Li *et al*, 2009). PCR duplicates and unmapped reads were removed, and the resulting BAM files were re-sorted and indexed. For visualisation in IGV, bigwig files were generated using bamCoverage (deepTools2). Biological replicates were merged using SAMtools, and peaks were called using MACS2 (Feng *et al*, 2012) after downsampling the input files to the size of the H3K27ac files. MACS peaks were used for super-enhancer identification using Rank Ordering of Super-Enhancers (ROSE; Whyte *et al*, 2013) with a stitching distance of 12,500 bp and a promoter exclusion zone ± 2,000 bp. For the minor allele homozygote samples, activity at the super-enhancer locus was calculated by summing the activity of detected enhancers within the region and normalising for region size.

### *In silico* transcription factor binding analysis

Transcription factor binding motifs enriched at constituent elements within the 6q23 super-enhancer were identified using TRAP (multiple sequences; Thomas-Chollier *et al*, 2011), with all human

promoters as the reference and an FDR correction for multiple testing (Benjamini & Hochberg, 1995). Motifs were obtained from the JASPAR CORE vertebrate database. Pathway analysis of enriched transcription factors within KEGG pathways was performed using g: Profiler (https://biit.cs.ut.ee/gprofiler/gost) with an FDR correction (Benjamini & Hochberg, 1995).

## Promoter capture Hi-C analysis

Interactions of the 6q23 super-enhancer in stimulated CD4 T cells were identified from existing promoter capture Hi-C data (Javierre et al, 2016) using the capture Hi-C plotter (https://www.chicp.org).

## qPCR in CD4 T cells from inflammatory bowel disease patients

131 patients with active IBD were recruited before treatment (Lee et al, 2011; Biasci et al, 2019). All patients provided written informed consent, and ethical approval was provided by the Cambridgeshire Regional Ethics committee (REC:08/H0306/21). Experiments conformed to the principles set out in the WMA Declaration of Helsinki and the Department of Health and Human Services Belmont Report. CD4 T cells were positively selected from a 100 ml blood sample, as described earlier. Cells were immediately lysed, and RNA and DNA were extracted (AllPrep DNA/RNA Mini kit, Qiagen). Genotyping was performed using the Illumina HumanOmniExpress 12v1.0 BeadChip, and data were processed as previously described (Peters et al, 2016). qPCR for genes at the 6q23 locus was performed in triplicate on a Roche LightCycler 480, using exon-spanning primers, with beta-actin as a reference (QuantiFast SYBR Green PCR Kit; Qiagen).

## CRISPR-Cas9 editing in resting CD4 T cells

### Guide RNAs
gRNA sequences are provided in Appendix Table S3.

### Optimisation
For RNP-based CRISPR editing in resting CD4 T cells, the following conditions were tested: nucleofection buffer (Human T cell Nucleofector kit, Lonza; "1 M" nucleofection solution), program (U014; V024) and use of electroporation enhancer (IDT). Positive control gRNAs: *HPRT*, *CXCR4*. Non-targeting controls: Alt-R CRISPR-Cas9 Negative Control crRNA #1 and #3 (IDT). To generate functional sgRNA duplexes, crRNAs and tracrRNA (Alt-R CRISPR-Cas9 tracrRNA, ATTO™ 550) were reconstituted in duplex buffer (200 μM), mixed 1:1 and heated to 95°C for 10 min before slowly cooling. Cas9 RNPs were generated immediately before use by adding Cas9 (Alt-R S.p. HiFi Cas9 Nuclease V3, 61 μM; IDT) to the gRNA duplex in a 1:3 ratio and incubating at 37°C for 20 min. 5 μl Cas9 RNP was nucleofected into 1 M CD4 T cells (positively selected from fresh single leucocyte cones; National Blood Service, Cambridge, UK) in 100 μl nucleofection buffer (program V024). Electroporation enhancer was added where indicated (final concentration 4 μM). After nucleofection, 500 μl pre-warmed media was added and cells were transferred to a 24-well flat-bottomed plate (total volume 1 ml/well) and cultured at 37°C, 5% $CO_2$. Cell culture media: X-VIVO15 (STEMCELL), 5% FBS, 50 μM 2-mercaptoethanol, and 10 μM *N*-acetyl L-cystine. After 6 h, the media was changed and

fresh pre-warmed media containing low dose IL-7 (1 ng/ml; PeproTech) was added to promote T-cell survival without stimulation. 48 h after nucleofection, cells were stimulated with anti-Biotin MACSiBead Particles loaded with CD2/3/28 antibodies (bead-to-cell ratio 1:2, Miltenyi) and IL-2 (10 ng/ml). Cells were harvested 24 h after stimulation and used for flow cytometry or lysed. Surface expression of CXCR4 was assessed by flow cytometry (stained in 250 μl): CXCR4 APC (1:100, #306510, BioLegend), Zombie Aqua Fixable Viability Kit (Biolegend) and Fc receptor blocking reagent (Miltenyi). Viability was ~ 80% at the end of the experiment.

### Editing efficiency assessment
DNA and RNA were extracted from cell lysates as described. For optimisation experiments, editing efficiency was estimated using a T7 Endonuclease assay (IDT). An additional Proteinase K incubation was used to inactivate T7 endonuclease I before fragment analysis. Digested heteroduplexes were quantified using a Bioanalyzer 2100 (Agilent). The optimal conditions for RNP nucleofection in resting CD4 T cells were 1 M buffer and V024 program, with electroporation enhancer. These conditions were used for subsequent experiments, in which editing efficiency was estimated using ICE (Inference of CRISPR Edits, Synthego; preprint: Hsiau et al, 2019)— a method that mathematically infers the composition of indels using Sanger sequencing traces from edited and non-edited samples.

### Deletion of NF-κB binding site at rs6927172 locus
gRNAs flanking the rs6927172-containing NF-κB binding site were designed (http://crispr.mit.edu) and checked for suitable on- and off-target activity (GPP sgRNA design tool: https://portals.broadinstitute.org/gpp/public/analysis-tools/sgrna-design). Two gRNAs proximal (5′) to the NF-κB motif (termed D and F) and two distal (3′) gRNAs (termed B and H) were synthesised (IDT). To reduce the chance that an effect might be due to off-target activity, RNPs were used in different combinations (one 5′ gRNA RNP with one 3′ gRNA RNP). Predicted indels: DB, 33 bp; DH, 50 bp; FB, 18 bp; FH, 35 bp. 5 μl Cas9 RNP (either equimolar amounts of 5′ and 3′ gRNA-containing RNPs or a non-targeting control) were nucleofected with 1 μl electroporation enhancer into resting CD4 T cells as described. For nascent RNA capture experiments, 5-ethynyl uridine (EU, Click-iT™ Nascent RNA Capture Kit, Thermo Fisher Scientific) was added at the time of stimulation (final concentration 0.4 mM). 24 h later, the supernatant was frozen for cytokine analysis, and the cells were harvested for flow cytometry or lysed. Six biological replicates were performed, although the RNP combination (FB) with very low editing efficiency was not repeated after the first two replicates.

### Deletion of other candidate SNPs in the 6q23 super-enhancer
gRNAs flanking other candidate SNPs within the 6q23 super-enhancer were designed, synthesised and incorporated into gRNA-Cas9 RNPs as described earlier. An equimolar mix of 5′ and 3′ RNPs were nucleofected into resting CD4 T cells, which were stimulated 48 h later and EU added, as described. After 24 h, cells were harvested, and DNA and RNA were extracted.

### Editing of TNFAIP3
Two gRNAs targeting *TNFAIP3* were obtained from the Brunello genome-wide CRISPR library (Doench et al, 2016) and synthesised (IDT). RNPs for each gRNA were generated and nucleofected into

## The paper explained

### Problem

Genome-wide association studies (GWAS) have identified hundreds of regions of the human genome that are involved in the pathogenesis of immune-mediated diseases, but the mechanisms by which these loci drive disease remain largely unknown. Identifying causal variants, whose biological effects mediate disease risk, is an important first step but fine-mapping efforts have been frustrated by strong linkage disequilibrium (LD), leaving most loci unresolved and the potential of GWAS unfulfilled.

### Results

This study describes the development of an adapted massively parallel reporter assay for use in primary CD4 T cells—the cell type whose regulatory DNA is most enriched for immune-mediated disease SNPs. By using this method to study gene deserts linked to a range of immune-mediated diseases, we simultaneously resolve the functional consequences of hundreds of non-coding candidate SNPs and show how this can identify putative causal variants via their functional effects. We illustrate the power of this approach using a locus linked to six autoimmune diseases that cannot be fine-mapped. By investigating the lead expression-modulating SNP, we uncovered an NF-κB-driven regulatory circuit which constrains T-cell activation through the dynamic formation of a super-enhancer that upregulates *TNFAIP3* (A20), a key NF-κB inhibitor. In activated T cells, this circuit is disrupted—and super-enhancer formation prevented—by the risk variant at the lead SNP, thus uncovering a mechanism that appears to broadly predispose to human autoimmunity.

### Impact

Our work provides a generalisable method that can distil disease-associated haplotypes down to specific functional variants, via their biological effects in disease-relevant primary cells. This generates testable hypotheses into disease mechanisms and could be applied to any non-coding association that overlaps regulatory elements in T cells, including those that cannot be fine-mapped. This can provide insights into disease biology, and a framework by which the considerable potential of GWAS in immune-mediated disease could finally be realised.

resting CD4 T cells. After 48 h, cells were stimulated, and 24 h later, the supernatant was frozen for cytokine analysis, and the cells harvested for flow cytometry.

### Flow cytometry

Surface staining was performed in 250 μl using CD69 BV421 antibody (1:100, #310930, BioLegend), Zombie Aqua Fixable Viability Kit (Biolegend) and Fc receptor blocking reagent (Miltenyi). ATTO-550 staining (ATTO-550-conjugated tracrRNA) was used to identify cells containing the RNP. Intracellular staining (for experiments in which the rs6927172-containing NF-κB binding site was deleted) was performed using the eBioscience Foxp3 Staining kit (Thermo Fisher) and a Phospho-IkB alpha (Ser32, Ser36) eFluor660 antibody (1:50, #12-9035-42, Thermo Fisher) with a fluorescence-minus-one control.

### Nascent RNA capture

Following RNA extraction, EU-labelled RNA was biotinylated and purified using Dynabeads Streptavidin T1 magnetic beads (Click-iT Nascent RNA Capture Kit, Life Technologies). Reverse transcription was performed using bead-bound RNA (SuperScript VILO cDNA synthesis kit), and qPCR was performed in triplicate (QuantiFast SYBR Green; Qiagen) on a Roche LightCycler 480 with beta-actin as reference gene. Expression of target genes was normalised to the non-targeting control.

### Cytokine quantification

For experiments in which the rs6927172-containing NF-κB binding site was deleted, supernatant cytokines were quantified in duplicate by electrochemiluminescence (MesoScale Discovery Immunoassay). For experiments in which *TNFAIP3* was directly edited, cytokines were quantified in triplicate using Quantikine ELISAs (R&D).

### Statistical methods

Statistical methods used in MPRA analysis are described above. For other analyses, comparison of continuous variables between two groups was performed using a paired *t*-test or one-sample *t*-test when comparing against a hypothetical value. A Shapiro–Wilk test was used to confirm normality, and sample sizes were based on having 80% power to detect a standardised effect size of 2, $\alpha = 0.05$ (and corrected for multiple testing as described). Two-tailed tests were used as standard unless a specific hypothesis was being tested.

## Data availability

The data sets produced in this study, including raw and processed files, are available in the following databases:

- MPRA data: Gene Expression Omnibus GSE135925 (https://www.ncbi.nlm.nih.gov/geo/query/acc.cgi?acc = GSE135925).
- ChIP-Seq data: Gene Expression Omnibus GSE136092 (https://www.ncbi.nlm.nih.gov/geo/query/acc.cgi?acc = GSE136092).

**Expanded View** for this article is available online.

## Acknowledgements

We acknowledge J.Lewandowski, M.Parkes, A.Kaser, D.Thomas and members of the Smith and Rinn labs for helpful discussion, J.Sowerby for experimental help and D.Seyres for ChIP advice. Jurkats were a gift from Stefan Marciniak, pRSCgfp-hAIM2 was a gift from Emad Alnemri (Addgene #51666), CBFRE-EGFP was a gift from Nicholas Gaiano (Addgene #17705), and pOTTC407-pAAV EF1a eGFP was a gift from Brandon Harvey (Addgene #60058). We thank NIHR BioResource volunteers for their participation, and acknowledge NIHR BioResource centres, NHS Blood and Transplant, and NHS staff for their contribution. This work was supported by the Wellcome Trust (Intermediate Clinical Fellowship to J.C.L, 105920/Z/14/Z; Senior Fellowship to C.W., WT107881), Crohn's and Colitis UK (M2018/3), the National Institute for Health Research (Cambridge Biomedical Research Centre at the Cambridge University Hospitals NHS Foundation Trust), the Howard Hughes Medical Institute (Gilliam Fellowship to A.G.), the Engineering and Physical Sciences Research Council & GlaxoSmithKline (iCase studentship to A.H., EP/R511870/1), the Medical Research Council (MC UU 00002/4 to C.W.) and the National Institutes of Health Oxford-Cambridge Scholars Program. The views expressed are those of the authors and not necessarily those of the NIH, the NHS, the NIHR or the Department of Health and Social Care.

## Author contributions

Conceptualization, JLR and JCL; Methodology, CB, AFG, OSB, and JCL; Software, AFG and KM; Investigation CB, AFG, CG, AH, TH, TA, MWE, and JCL; Writing – Original Draft, JCL; Writing – Review & Editing, all authors; Funding Acquisition, JCL; Supervision, CW, KGCS, JLR, and JCL.

## Conflict of interest

The authors declare that they have no conflict of interest.

## For more information

TNFAIP3 (OMIM): https://www.omim.org/entry/191163
GWAS catalog: https://www.ebi.ac.uk/gwas/
Author website: https://www.citiid.cam.ac.uk/james-lee/

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
