## [Review Process File · EMBO Molecular Medicine]

Resolving mechanisms of immune-mediated disease in primary CD4 T cells

Bourges C, Groff AF, Burren OS, Gerhardinger C, Mattioli K, Hutchinson A, Hu T, Anand T, Epping MW, Wallace C, Smith KGC, Rinn JL, Lee JC

Review timeline:

Submission date:	1st Feb 2020
Editorial Decision:	17th Feb 2020
Revision received:	4th Mar 2020
Accepted:	9th Mar 2020

Editor: Céline Carret

Transaction Report:

(Note: An earlier version of this manuscript was assessed by another journal and was then transferred to EMBO Molecular Medicine. As the original review of the manuscript was performed outside of EMBO Molecular Medicine's transparent review process policy, no Peer Review Process information is available for this article. With the exception of the correction of typographical or spelling errors that could be a source of ambiguity, letters and reports are not edited. The original formatting of letters and referee reports may not be reflected in this compilation.)

1st Editorial Decision

17th Feb 2020

Thank you for the submission of your manuscript to EMBO Molecular Medicine. I am happy to report that we have now received the enclosed recommendations from our editorial advisers.

As you will see the advisers are supportive of publication and I am pleased to inform you that we will be able to accept your manuscript pending the following final amendments:

1) Please address the minor changes commented by advisors #1 and #2. We would like to encourage you to provide the data suggested and modify figures and text as recommended by this advisor.

Please provide a point-by-point letter INCLUDING my comments as well as the advisor's reports and your detailed responses to their comments (as Word file).

***** Reviewer's comments *****

Advisor #1:

I had a look to the Ms and authors' reply to reviewers' critics. In my opinion, this Ms merits publication since by using a seemingly new approach they show that it is possible to provide biologically relevant information related to non coding variants shown to affect a superenhancer at least in one proof of principle case.

It is a quite difficult Ms to read, though now improved, perhaps not as strong in message as the authors claim (i.e., to potentially decipher the meaning of potentially all non coding variants), i.e. some rewriting is needed to make the paper more accessible to the general readers and maybe tuning down the strong claims of the abstract. But still, as said above

this approach adds to the armamentarium of potential strategies to tackle such (tough) questions.

Thus I am inclining to be positive.

Advisor #2:

I had the opportunity to go over the manuscript and the response to reviewers. I think the authors have clarified all the points raised by the reviewers. I think the manuscript will be of substantial interest to the broader complex genetics/functional genomics community. The experiments are well presented and the results advance our knowledge, especially in the context of functional consequences at the TNFAIP3 locus. I enjoyed reading this manuscript and thought the results were very relevant.

I have a few comments that I think would make the manuscript stronger if the authors would be willing to address them (I appreciate that they have already done comprehensive work addressing the previous comments).

1) Overall, I think the approach is important and I was disappointed that the authors didn't provide more comprehensive comparison of MPRA results over other approaches that prioritise functional variants, e.g. they briefly mention conservation scores in the context of the NFkB binding at TNFAIP3 locus, similar analysis could be done across all the tested variants to provide an overview of how many variants could have been prioritised prior the MPRA approach.

Somewhat on this note, I didn't have the access to supplementary materials, I think one piece of information that would be critical to share with the community alongside the publication is a supplementary table with all the variants tested, the primer sequences, raw result values from their experiments and computed expression values. I can imagine many researchers might be interested to check if the loci they are working have been tested in this experiments and with what results.

2) In response to the Reviewer 1 comments [about performing multiple comparisons using MPRA in primary cells vs. cell lines]:

I agree that the multiple comparisons are beyond the scope of the manuscript. However, I also agree with the initial intuition of the authors to include the comparison between Jurkat and T cells. Even if well established, this is an important message and it is critical we think about ways to follow up GWAS signals in the relevant cellular models. Therefore, I suggest including the comparison panel from Figure 2 in the manuscript, perhaps as a part of a supplementary figure.

3) In figures, the pie charts are used to depict posterior probabilities of fine-mapped SNPs and they are not a good representation, the posterior probability values are not directly depicted and difficult to infer from the pie charts themselves, I suggest a different representation or providing the exact values.

1st Revision - authors' response

4th Mar 2020

**** Reviewer's comments ****

Advisor #1:

I had a look to the Ms and authors' reply to reviewers' critics.
In my opinion, this Ms merits publication since by using a seemingly new approach they show that it is possible to provide biologically relevant information related to non coding variants shown to affect a superenhancer at least in one proof of principle case.

It is a quite difficult Ms to read, though now improved, perhaps not as strong in message as the authors claim (i.e., to potentially decipher the meaning of potentially all non coding variants), i.e. some rewriting is needed to make the paper more accessible to the general readers and maybe tuning down the strong claims of the abstract. But still, as said above this approach adds to the armamentarium of potential strategies to tackle such (tough) questions.

Thus I am inclining to be positive.

We are grateful for these constructive comments. We did not intend to imply that adapted MPRA could decipher the meaning of all non-coding variants, and have now re-written the abstract in line with the advisor's suggestions. We have also rewritten some of the other sections that we think may not have been fully accessible to general readers. Edits are tracked in the revised version of the manuscript.

Abstract

“Deriving mechanisms of immune-mediated disease from GWAS data remains a formidable challenge, with attempts to identify causal variants being frequently hampered by strong linkage disequilibrium. To determine whether causal variants could be identified from their functional effects, we adapted a massively-parallel reporter assay for use in primary CD4 T-cells, the cell-type whose regulatory DNA is most enriched for immune-mediated disease SNPs. This enabled the effects of candidate SNPs to be examined in a relevant cellular context, and generated testable hypotheses into disease mechanisms. To illustrate the power of this approach, we investigated a locus that has been linked to 6 immune-mediated diseases but cannot be fine-mapped. By studying the lead expression-modulating SNP, we uncovered an NF- κ B-driven regulatory circuit which constrains T-cell activation through the dynamic formation of a super-enhancer that upregulates TNFAIP3 (A20), a key NF- κ B inhibitor. In activated T-cells, this feedback circuit is disrupted – and super-enhancer formation prevented – by the risk variant at the lead SNP, leading to unrestrained T-cell activation via a molecular mechanism that appears to broadly predispose to human autoimmunity.”

Advisor #2:

I had the opportunity to go over the manuscript and the response to reviewers. I think the authors have clarified all the points raised by the reviewers. I think the manuscript will be of substantial interest to the broader complex genetics/functional genomics community. The experiments are well presented and the results advance our knowledge, especially in the context of functional consequences at the TNFAIP3 locus. I enjoyed reading this manuscript and thought the results were very relevant.

I have a few comments that I think would make the manuscript stronger if the authors would be willing to address them (I appreciate that they have already done comprehensive work addressing the previous comments).

We are grateful for these positive comments and for the time the advisor has taken in reading our manuscript and making constructive suggestions.

1) Overall, I think the approach is important and I was disappointed that the authors didn't provide more comprehensive comparison of MPRA results over other approaches that prioritise functional variants, e.g. they briefly mention conservation scores in the context of the NF κ B binding at TNFAIP3 locus, similar analysis could be done across all the tested variants to provide an overview of how many variants could have been prioritised prior the MPRA approach.

We have now used 2 *in silico* methods to assess how many of these variants could have been prioritised by other means (DeepSEA, which uses evolutionary conservation and

predicted chromatin effects, and RegulomeDB, which uses a variety of public datasets). The results of these analyses are presented in a new Expanded View Dataset (Dataset EV1). Between these 2 methods, the lead MPRA SNP was classified as the most significant functional SNP at 3 of the 14 loci (2 by DeepSEA and 1 by RegulomeDB). DeepSEA also predicted that 3 other lead SNPs would be functionally significant, but ranked other candidate SNPs ahead of them (most of which had no expression-modulating effect in the MPRA). These *in silico* methods have been reported to be better at predicting negative effects compared with positive effects, and consistent with this, rs1736137 was included in the SNPs classified as non-functional. This is the variant shown in Fig 3A, which was previously identified as a causal variant by fine-mapping (and which has a highly significant expression-modulating effect in MPRA). Overall, these data indicate that while other methods can sometimes identify true expression-modulating variants, they are often misleading and cannot substitute for experimental methods. We have added the following text to the Results:

“To determine whether these variants could have been prioritised by other means, we compared the MPRA results with *in silico* methods designed to identify functional variants – DeepSEA (Zhou & Troyanskaya, 2015) and RegulomeDB (Dong & Boyle, 2019) (Dataset EV1). Considering these approaches together, the lead MPRA SNP was predicted to be the most functionally significant variant at 3/14 loci (2 by DeepSEA, 1 by RegulomeDB). DeepSEA also predicted that 3 more lead SNPs would be functionally significant, but prioritised other candidate SNPs at these loci (most of which had no expression-modulating effect in CD4 T cells). At the remaining 8 loci, the lead MPRA SNP was not predicted to have an expression-modulating effect – consistent with these methods being better at predicting negative effects than positive effects (Dong & Boyle, 2019) and highlighting the value of studying disease-associated loci in relevant primary cells.”

Somewhat on this note, I didn't have the access to supplementary materials, I think one piece of information that would be critical to share with the community alongside the publication is a supplementary table with all the variants tested, the primer sequences, raw result values from their experiments and computed expression values. I can imagine many researchers might be interested to check if the loci they are working have been tested in this experiments and with what results.

Much of this data was already included in the supplementary tables (now Datasets EV2 and EV3) which represent the meta-analysis results for every variant tested in resting and stimulated CD4 T cells respectively. This will enable researchers to examine the results at their loci of interest. The raw and processed sequencing values, and all of the sequences tested, are provided in our GEO submission (GSE135925). We can also provide these files separately (e.g. as additional Extended View Datasets) if this is felt to be necessary, but they are very large files, and it would probably be easier for interested researchers to access them from GEO. Let us know what you would prefer.

2) In response to the Reviewer 1 comments [about performing multiple comparisons using MPRA in primary cells vs. cell lines]:

I agree that the multiple comparisons are beyond the scope of the manuscript. However, I also agree with the initial intuition of the authors to include the comparison between Jurkat and T cells. Even if well established, this is an important message and it is critical we think about ways to follow up GWAS signals in the relevant cellular models. Therefore, I suggest including the comparison panel from Figure 2 in the manuscript, perhaps as a part of a supplementary figure.

We very much appreciate this comment, and agree that this is an important message. We have now included the comparison panel in Appendix Figure S1 and added text to explain this:

“The effects observed in resting and stimulated CD4 T cells were highly correlated (Appendix Fig S1E), but these effects did not correlate particularly well with results

obtained in Jurkat cells (an immortalised CD4 T cell line) – reinforcing the value of using an appropriate cellular model when studying human disease (Appendix Fig S1F).”

3) In figures, the pie charts are used to depict posterior probabilities of fine-mapped SNPs and they are not a good representation, the posterior probability values are not directly depicted and difficult to infer from the pie charts themselves, I suggest a different representation or providing the exact values.

We have now added the exact posterior probability values to these figure panels.

The authors performed the requested editorial changes.

Corresponding Author Name: James Lee
Journal Submitted to: EMBO Molecular Medicine
Manuscript Number: EMM-2020-12112